# Heteroalleles in Common Wheat: Multiple Differences between Allelic Variants of the *Gli-B1* Locus

**DOI:** 10.3390/ijms22041832

**Published:** 2021-02-12

**Authors:** Eugene Metakovsky, Laura Pascual, Patrizia Vaccino, Viktor Melnik, Marta Rodriguez-Quijano, Yulia Popovych, Sabina Chebotar, William John Rogers

**Affiliations:** 1Department of Biotechnology-Plant Biology, School of Agricultural, Food and Biosystems Engineering, Universidad Politécnica de Madrid, 28040 Madrid, Spain; emetakovsky@gmail.com (E.M.); marta.rurquiaga@upm.es (M.R.-Q.); 2Consiglio per la Recerca in Agricultura e l’Analisi dell’Economia Agraria, Research Centre for Cereal and Industrial Crops, 13100 Vercelli, Italy; patrizia.vaccino@crea.gov.it; 3Vavilov Institute of General Genetics RAS, 117971 Moscow, Russia; meller@vigg.ru; 4Department of Genetics and Molecular Biology, National I.I. Mechnikov University, 65058 Odessa, Ukraine; popovych1818@gmail.com (Y.P.); s.v.chebotar@onu.edu.ua (S.C.); 5Departamento de Biología Aplicada, CIISAS, CIC-BIOLAB, CONICET-INBIOTEC, CRESCA, Facultad de Agronomía, Universidad Nacional del Centro de la Provincia Buenos Aires, 7300 Azul, Provincia de Buenos Aires, Argentina; rogers@faa.unicen.edu.ar

**Keywords:** *Triticum aestivum*, *Gli-B1*, γ-gliadin polymorphism, APAGE, PCR, DNA sequencing

## Abstract

The *Gli-B1*-encoded γ-gliadins and non-coding γ-gliadin DNA sequences for 15 different alleles of common wheat have been compared using seven tests: electrophoretic mobility (EM) and molecular weight (MW) of the encoded major γ-gliadin, restriction fragment length polymorphism patterns (RFLPs) (three different markers), *Gli-B1*-γ-gliadin-pseudogene known SNP markers (Single nucleotide polymorphisms) and sequencing the pseudogene *GAG56B*. It was discovered that encoded γ-gliadins, with contrasting EM, had similar MWs. However, seven allelic variants (designated from I to VII) differed among them in the other six tests: I (alleles *Gli-B1i*, *k*, *m*, *o*), II (*Gli-B1n*, *q*, *s*), III (*Gli-B1b*), IV (*Gli-B1e*, *f*, *g*), V (*Gli-B1h*), VI (*Gli-B1d*) and VII (*Gli-B1a*). Allele *Gli-B1c* (variant VIII) was identical to the alleles from group IV in four of the tests. Some tests might show a fine difference between alleles belonging to the same variant. Our results attest in favor of the independent origin of at least seven variants at the *Gli-B1* locus that might originate from deeply diverged genotypes of the donor(s) of the B genome in hexaploid wheat and therefore might be called “heteroallelic”. The donor’s particularities at the *Gli-B1* locus might be conserved since that time and decisively contribute to the current high genetic diversity of common wheat.

## 1. Introduction

Wheat is globally one of the most important crops. Currently, with world production at 728.5 million tonnes and a sown area covering 214 million hectares in 2019, wheat includes a quarter of total cereal production [1]. Wheat production includes two different polyploidy species derived from natural hybridization events [2], the tetraploid species (AABB) durum wheat (*Triticum turgidum L.*) and the hexaploid species (AABBDD) bread wheat (*Triticum aestivum L.*). Bread wheat represents roughly 90 to 95% of the total production; hence, any advances related to knowledge or breeding in this species will have a major worldwide impact.

Wheat provides 20% of the daily protein and food calorie intakes worldwide and represents one of the major protein components of the human diet. In recent decades, there has been an unprecedented increase in knowledge of the chemistry, genetics and functionality of the main proteins of the wheat seed, gliadin and glutenin. For example, there are classifications of different alleles at the seed-storage-protein-encoding loci in their relation to dough quality. This knowledge helps the production of new genotypes with improved characteristics (including field, dough-quality and medical aspects). However, the immense complexity of the bread wheat genome (2n = 6x = 42, 17 Gb) and of the genetic loci encoding the synthesis of the gliadins and glutenins leaves many questions unanswered. Such as the fact the same protein alleles might have different relationships to dough quality. Such findings raise the question of whether these alleles, identified by protein analysis in unrelated cultivars, are indeed identical. Moreover, the degree of difference at the DNA level of allelic variants differing on the protein level at the same seed-storage-protein-encoding locus is unknown.

A common wheat genotype produces more than 50 different polypeptides of gliadin (gliadins) [3,4]. The primary structure of several hundred gliadin-coding DNA sequences from different wheat species has been studied, and three main types of gliadin polypeptides differing between them in primary structure and biochemical properties, the so-called α-, γ- and ω-gliadins, have been described [5]. Moreover, each type, for example, the γ-gliadins [6,7], is composed of a number of families, meaning that members of a given family are more similar than members of different families.

About 20–25 bands differing in electrophoretic mobility (EM) may be observed in a one-dimensional electrophoregram (acid polyacrylamide gel electrophoresis, APAGE) of a single grain, encoded in common wheat by six major (*Gli-A1*, *Gli-B1*, *Gli-D1*, *Gli-A2*, *Gli-B2*, *Gli-D2*) [8] and several minor [9] *Gli* loci. Analysis of segregating progenies from crosses between wheat genotypes showed that any *Gli* allele encodes two or more gliadin polypeptides (a block of electrophoretic bands) inherited together as a Mendelian unit. Alleles at one *Gli* locus differ in the number and/or EM of the encoded gliadin polypeptides of different gliadin types [10]. In total, there are currently about 180 known alleles at the six major loci [11].

A fundamental characteristic of the seed-storage-protein-coding loci (for example, of the tightly linked *Gli-1* and *Glu-3* loci mapped in the distal part of the short arms of the homoeologous group 1 chromosomes [8]), is their great complexity. The size of one storage protein gene, including its regulatory sequences, is known to be about 0.8–1.5 Kb [12,13,14], but the distance between these genes has been shown to be very large, reaching 55–120 Kb per 1 cM of genetic distance [15,16]. The average distance between active genes was about 81 Kb [14], and the distance of 10 Kb between two gliadin genes, tenfold the length of a gliadin gene itself, may be considered as exceptionally short, besides the fact that the genes can be considered to be clustered [16]. The size of an entire *Gli* locus encoding several gliadin polypeptides may exceed 2000 Kb as suggested for the *Gli-A2* locus [17].

Hence all expressed gliadin genes together occupy only a minor part of the DNA of a *Gli* locus. The largest part of a *Gli* locus was shown to consist of non-coding DNA sequences represented mainly by different types of transposable elements [16,18,19], but also by gliadin pseudogenes and gene fragments [20] for which inter-varietal polymorphism was described [21] and even some non-gliadin genes and sequences with open reading frames [16,22,23]. Therefore, allelic variants of a *Gli* locus may differ in the number and types of active genes, in the distance between them, in the quota of pseudogenes among all the gliadin sequences and, probably, in the presence of non-gliadin genes harbored within a given *Gli* locus. The polymorphism of an entire *Gli* locus could be further augmented by the presence of single nucleotide polymorphisms (SNP).

Alleles at a *Gli* locus are routinely identified by the use of APAGE of gliadin proteins, thereby revealing sets of transcribed genes within the locus. However, an allele identified by APAGE may differ, in different genotypes, in the non-coding part of the locus. Hence alleles identified by APAGE may only be considered as markers of corresponding allelic variants of *Gli* loci.

There are few possibilities to compare allelic variants of an entire *Gli* locus, including its non-coding sequences. Restriction fragment length polymorphism (RFLP) analysis may disclose a difference between the alleles in the distribution of gliadin-coding sequences (including pseudogenes) along the whole locus. Two sets of molecular probes were also made available for the analysis of gliadin-coding DNA sequences. Firstly, the *GAG56B* γ-gliadin pseudogene was detected with the gene-specific PCR primer pair GAG29/30, and one of the variants of the pseudogene sequence (p-aes) was proven to be a characteristic of common wheat *T. aestivum* [24]. Secondly, two allele-specific PCR markers were developed for the *Gli-B1*-γ-gliadin sequence (probable pseudogene). Australian common wheat cultivars tested with these markers gave one PCR product with one pair of primers of either 369 or 397 bp [25].

Thus, a *Gli*- locus is a highly complex chromosomic fragment that includes several gliadin-encoding genes, pseudogenes and sequences unrelated to gliadins. Some allelic variants at the *Gli* loci (and at the *Glu-3* loci tightly linked to *Gli-1*) have been related to dough quality, providing useful markers to conduct quality breeding. However, the same allele at the *Gli* locus identified by protein electrophoresis in different cultivars might apparently have different relationships to dough quality, suggesting that this allele might be represented by more than one variant and tampering with the usefulness of gliadins as molecular markers. In order to explore this question, in this work, we compared 15 allelic variants at the *Gli-B1* locus of common wheat (identified as blocks of *Gli-B1*-encoded gliadin polypeptides in APAGE) by means of several approaches: firstly, EM and molecular weight (MW) of encoded γ-gliadins using two-dimensional electrophoresis; secondly, their RFLP patterns (three different restriction enzymes and a γ-gliadin-specific probe); thirdly, the length of the PCR sequence of the γ-gliadin pseudogene; and fourthly, the DNA sequence of another γ-gliadin pseudogene present within the *Gli-B1* locus. It was shown that the alleles studied might be divided into several variants strongly differing between them in their coding, as well as non-coding, sequences. The simultaneous multiple differences (without intermediate forms) among versions of the *Gli-B1* locus may be explained through the suggestion that these variants maintain the particularities of the original genotypes of the corresponding diploid donor(s) in hexaploid wheat.

## 2. Results

### 2.1. Alleles at the Gli-B1 Locus Identified by APAGE

Twenty-four alleles at the *Gli-B1* locus (including the null-allele and the *Gli-B1l* allele which designates the 1BL.1RS translocation) have been revealed in the wheat germplasm studied [11], and 16 of them, including *Gli-B1l*, were selected for further analysis.

Examples of gliadin electrophoregrams (APAGE) of the common wheat cultivars carrying the alleles studied (Figure 1) and schemes of blocks of jointly inherited gliadin bands (Appendix A) are shown.

Alleles producing the slowest-moving γ-gliadin and alleles *Gli-B1n*, *q*, *s* and *h* are commonly accompanied by the allele *Gli-B5b* of the *Gli-B5* locus, tightly linked to *Gli-B1* [26]. The allele *Gli-B5b* encodes a pair of faint ω-gliadins, while the allele *Gli-B5a* is null. Allelic variation at *Gli-B5* was accepted as a polymorphism procured by the distal part of chromosome 1B, and ω-gliadins encoded by the *Gli-B5b* were considered as a part of the *Gli-B1*-encoded blocks [11].

Each allele at the *Gli-B1* locus of common wheat produces one γ-gliadin (the *Gli-B1a* encodes two γ-gliadins), two or more ω-gliadins and some bands in the β-zone of the APAGE electrophoregram (Appendix A). At least seven variants of the EM of the *Gli-B1*-encoded γ-gliadin (therefore, of the locus) were revealed: *Gli-B1i*, *m*, *k*, *o* (variant I, the slowest EM); *Gli-B1n*, *q*, *s* (II); *Gli-B1b* (III); *Gli-B1e*, *g* (IV); *Gli-B1h* (V); *Gli-B1d* (VI); and *Gli-B1a* (VII, the fastest) (Figure 1). The allele *Gli-B1f* produces a slightly faster γ-gliadin compared to alleles of variant IV (Figure 1l,j,k), but nevertheless may be attributed to this variant due to the similarity of other APAGE bands composing a block. For the same reason (the composition of the encoded block), the allele *Gli-B1b* was assumed to be different from alleles of the group IV.

The allele *Gli-B1c* produces a weaker stained γ-gliadin and is linked to *Gli-B5b* (Figure 1i), thus differing from other alleles of group IV. Hence, the *Gli-B1c* might be considered as some kind of variation of variant IV or even a rather independent variant VIII of the *Gli-B1* locus.

### 2.2. Apparent Molecular Weight of the Gli-B1-Encoded γ-Gliadins as Studied by Two-Dimensional APAGE × SDS Electrophoresis

The best way to compare the apparent MWs of the *Gli-B1*-encoded γ-gliadin polypeptides using two-dimensional electrophoresis (TDE) is to analyze a heterozygous seed of a cross between two genotypes carrying the alleles under study. Additionally, there is an “internal marker” present in most of the genotypes, the *Gli-A3*-controlled ω-gliadin: allelic variants of this polypeptide show identical apparent MWs of about 41 kD [9]. The major γ-gliadin encoded by the *Gli-D1* locus (apparent MW of about 34 kD, data not shown) may be also used as an internal marker of MW (Figure 2 and Appendix A).

The intriguing result of this approach seems to be a close similarity (identity) of the apparent MWs of γ-gliadins produced by any variant of the *Gli-B1* locus (Figure 2 and Appendix A): *Gli-B1*-encoded γ-gliadins with different (including contrasting) EM had similar MWs of about 41 kD. For example, the MWs of γ-gliadins of alleles of variants I, III, VI (Figure 2a–c) and IV, II, V (Appendix A, respectively) were hardly distinguishable by TDE. The close similarity of the MWs of these γ-gliadins was confirmed by analysis of heterozygous F2 seeds (Figure 2d).

Analysis of TD (APAGE × SDS) electrophoregrams published by different authors, as well as the results of our work, showed that alleles *Gli-B1a*, *b*, *c*, *d*, *e*, *f*, *g*, *h*, *k*, *m*, *o*, *q* and *s* each controlled γ-gliadin of similar MWs (Appendix A). A new allele, *Gli-B1v* (differing slightly from the *Gli-B1o* in the EM of one ω-gliadin), found in Spanish landraces also produced the γ-gliadin with an MW identical to those of variant I (Appendix A). In durum wheat, known γ-gliadins γ42 and γ45 differing in their EM also have similar MWs of about 41 kD (Figure 3 in [27]).

Although SDS-electrophoresis is hardly sensitive enough to document some probable minor differences in the length of compared polypeptides (for example, Figure 2e), it is clear that γ-gliadins controlled by different alleles at the Gli-B1 locus have a similar (if not identical) MW. Therefore, they did not meet the regularity (higher EM, lower MW) established earlier for the Gli-A1- and the Gli-D1-encoded γ-gliadins [28].

A general structure for the γ-gliadin-encoding genes and polypeptides is known [5,6,7]. The number of repeats and the length of the microsatellite present in the γ-gliadin gene sequence, as well as deletions or insertions, may influence the length (the MW) of the encoded γ-gliadin polypeptide. In contrast, considerable differences in the EM without noticeable alterations in their MW observed in our work, for example, among γ-gliadins of alleles *Gli-B1m*, *b*, *d* (Figure 2a–c, respectively), may depend upon the number of charged amino acids present in these polypeptides of similar length.

### 2.3. Gli-B1 Alleles Represented by RFLP Patterns

Allelic variants of the *Gli-B1* locus were studied by the RFLP approach using three different restriction enzymes *Taq*I, *Rsa*I or *Hae*III and the K32 probe, specific for the transcribed γ-gliadin sequences. Previous analysis of aneuploid lines of the cultivar Chinese Spring permitted identification of polymorphic DNA fragments belonging to the allele *Gli-B1a* in the RFLP pattern of this cultivar, and a congruity was established between the presence of definite polymorphic DNA fragments and some *Gli-B1* alleles in wheat genotypes [29].

The RFLP-*Taq*I profiles (Appendix A) for 12 alleles at the *Gli-B1* locus were compared, and 19 polymorphic bands were considered. The tree of relationships among *Gli-B1* alleles drawn from their RFLP patterns (Figure 3) showed that the endonuclease *Taq*I did not distinguish between alleles *Gli-B1m* and *k*, *Gli-B1n* and *q*, and *Gli-B1e* and *g*. Sets of polymorphic bands of alleles *Gli-B1s* and *Gli-B1f* produced by *Taq*I were close to those of variants II and IV, respectively. For example, the difference between the *Gli-B1f* and other alleles of variant IV was in the position of only one RFLP fragment (Appendix A; lane 1, band 19, and lane 3, band 16, respectively). The RFLP-*Rsa*I profiles produced 25 polymorphic bands (data not shown). The tree of relationships among *Gli-B1* alleles (Figure 3b) was generally similar to that obtained using *Taq*I. However, in contrast to *Taq*I, alleles *Gli-B1b* and *Gli-B1d* had nothing in common when *Rsa*I was used (Figure 3a,b).

The RFLP-*Hae*III profiles produced only nine polymorphic bands, and the relationships among *Gli-B1* alleles were different from those obtained using two other endonucleases. The most important findings were, first, an identity of the allele *Gli-B1a* with allelic variant IV, and, second, a considerable difference of the allele *Gli-B1c* from this variant (Figure 3c).

The results obtained with the RFLP approach show considerable differences in DNA sequence (positions of restriction sites) along the *Gli-B1* locus among its alleles. There are six clear-cut allelic variants repeating those revealed by TDE, *Gli-B1k* and *m* (variant I), *Gli-B1n* and *q* (+*s*) (II), *Gli-B1b* (III), *Gli-B1e* and *g* (+*f*) (IV), *Gli-B1d* (VI), and *Gli-B1a* (VII). Alleles of different variants do not share or share only a few identical *Gli-B1*-derived polymorphic DNA fragments recognized by the γ-gliadin-specific probe. Furthermore, the approach has permitted some fine differences to be revealed between alleles belonging to the same variant.

In accordance with the RFLP analysis, the allele *Gli-B1c* may represent one more variant (VIII), related to, but different from, alleles of group IV, as has already been suggested on the basis of APAGE data. The allele *Gli-B1a* seems to be related to variant IV (endonucleases *Rsa*I and *Hae*III).

There was no convincing identity of any RFLP band produced by genotypes carrying the allele *Gli-B1l* (the 1BL.1RS) and any other allele studied.

### 2.4. Comparison of Alleles at the Gli-B1 Locus Using γ-Gliadin Pseudogene SNP Markers

γ-Gliadin SNP markers (PCR primers GliB1.1 and GliB1.2) are specific for a pseudogene located inside the *Gli-B1* locus. Each cultivar tested produced a PCR sequence with either primer GliB1.1 or GliB1.2, whereas cultivars carrying the 1BL.1RS translocation did not produce any amplification [25]. In our work, 16 alleles (including *Gli-B1l*) at the *Gli-B1* locus were analyzed by this test (Figure 4; Table 1), and four main observations were drawn.

Firstly, all alleles at the *Gli-B1* locus were divided into two groups, according to [25]. One group gave a PCR product with the GliB1.1 primer and included alleles of variants I (*Gli-B1i*, *k*, *m*, *o*), II (*Gli-B1n*, *q*, *s*) and III (*Gli-B1b*). Another group (PCR product with the GliB1.2) was composed of all other alleles (*Gli-B1a*, *d*, *h*, *c*, *e*, *f*, *g*) (Table 1). As expected, the cultivar Cartaya carrying the 1BL.1RS translocation (allele *Gli-B1l*) did not give a PCR product with any pair of primers.

Secondly, there was a polymorphism in the length of the PCR products (Product Length Polymorphism, PLP) (Figure 4, zone A). Three length variants were detected with the GliB1.1 primers (Figure 4a, lines 6–12) and four with GliB1.2 (Figure 4a, lines 1–5; Figure 4b, line 9). A polymorphism observed in zone B of the DNA electrophoregram (Figure 4) that included some PCR-derived DNA fragments of higher length was not considered.

Thirdly, an allele that is present in genotypes of different cultivars produced a PCR sequence of apparent identical length. For example, three cultivars from three countries with the allele *Gli-B1b* and four cultivars from four countries with the allele *Gli-B1d* produced PCR sequences of approximately 369 bp (GliB1.1) and 409 bp (GliB1.2), respectively (Table 1).

Fourthly, alleles encoding slow-moving γ-gliadin (*Gli-B1i*, *k*, *m*, *o*; Appendix A) gave a PCR product of identical length (Figure 4a, lines 10–12; Table 1). In addition, alleles of variant IV (*Gli-B1e*, *g*, *f*) and allele *Gli-B1c* each produced an apparently identical PCR sequence of 397 bp (Figure 4b, lines 4–7).

Therefore, there are seven groups of alleles identified in accordance with the PLP. These groups are identical to those revealed by the APAGE, TDE and RFLP approaches: alleles *Gli-B1i*, *k*, *m*, *o* (variant I); *Gli-B1n*, *q*, *s* (II); *Gli-B1b* (III); *Gli-B1e*, *g*, *f* (IV), *Gli-B1h* (V); *Gli-B1d* (VI); and *Gli-B1a* (VII). The allele *Gli-B1c* belongs, in this test, to variant IV.

### 2.5. DNA Sequence of the γ-Gliadin Pseudogene GAG56B in Allelic Variants of the Gli-B1 Locus

Seven cultivars studied for polymorphism in their γ-gliadin pseudogene *GAG56B* of the *Gli-B1* locus were divided into two groups by DNA sequence [24]. In the present work, von Büren’s primers GAG29 and GAG30 were used and the resulting products were sequenced. In total, 30 cultivars from 15 countries representing 15 alleles at the *Gli-B1* locus were compared (Table 2).

We were able to obtain a high-quality sequence between 170 bp from the forward primer and 800 bp for 29 cultivars. In Salmone, a high-quality sequence ranged from 200 bp to 800 bp (Appendix A). Aligning against the reference genebank accession M13712, we identified 14 positions with SNP, one of them (position 259) with three different possible nucleotides. Furthermore, the sequence included a single sequence repeat (SSR) CAA for which 10 possible variants of the length were detected. All these polymorphisms were combined in 11 different haplotypes (Figure 5; Table 2).

Four main observations have been drawn from these results.

Firstly, in most cases, any allele at the *Gli-B1* locus (identified through APAGE analysis of the set of gliadin polypeptides) occurring in different cultivars has an identical sequence for the γ-gliadin pseudogene *GAG56B*. For example, haplotype 2 was registered in two unrelated cultivars from two countries with allele *Gli-B1n*, haplotype 4 occurred in three cultivars with *Gli-B1c*, and haplotype 5 in three cultivars from three countries with *Gli-B1h* (Table 2). Therefore, there was a clear congruity of the polymorphisms in the expressed (*Gli-B1*-encoded γ-gliadin) and noncoding (γ-gliadin pseudogene) parts of the *Gli-B1* locus. The allele *Gli-B1c* belongs, in this test, to variant IV.

Secondly, 11 haplotypes of the pseudogene *GAG56B* were divided into seven deeply diverging groups similar to those revealed by other methods in our work: alleles *Gli-B1i*, *k*, *m*, *o* (variant I: haplotype 1); *Gli-B1n*, *q*, *s* (II: 2); *Gli-B1b* (III: 3); *Gli-B1e*, *g*, *f*, *c* (IV: 4); *Gli-B1h* (V: 5); *Gli-B1d* (VI: 6); and *Gli-B1a* (VII: 7) (Figure 5). The smallest differences were observed between alleles of groups II and III (substitution at position 182 and a minimal difference in the length of the SSR sequence) and of the groups IV and VII (a difference in the length of the SSR sequence). More differences were found between variants I, IV + VII and II + III, while the greatest difference was observed between variants I and V (substitution in 10 positions and in the length of the SSR) (Figure 6).

Thirdly, there was one case of a fine difference between alleles belonging to the same group (as has been found using the RFLP approach): the allele *Gli-B1m* differed slightly from other alleles of variant I by the length of its SSR sequence (Figure 5, Table 2).

Fourthly, there were three alleles (identified by APAGE) for which the *GAG56B* sequence differed among wheat genotypes: *Gli-B1m* (haplotypes 1, 8 and 9), *Gli-B1e* (4 and 11) and *Gli-B1f* (4 and 10) (Figure 6, Table 2). The difference might be caused by a mutational change in a given allelic variant of the *Gli-B1* locus causing the appearance of a new haplotype of the pseudogene studied. For example, an expansion of the CAA repeats present inside the *GAG56B* pseudogene converts allelic variant *Gli-B1e* (the cultivar Glenlea, haplotype 4, usual for alleles of variant IV) into the new variant of the same allele (the cultivar Saratovskaya-36, haplotype 11). Three variants of the allele *Gli-B1m* differed in the number of the CAA repeats (18, 17, 20), while the only substitution in position 348 distinguished haplotype 4 from 10 (Figure 5, Appendix A). Any allele at *Gli-B1* (for example, *Gli-B1m*), harboring different haplotypes of the *GAG56B* pseudogene in different cultivars, always produces the same set (a block) of encoded gliadins (data not shown).

Finally, a close similarity was noticed between two different trees of genetic distances, based, firstly, on the RFLP-*Rsa*I data (Figure 3b), and, secondly, on the sequencing of the *GAG56B* pseudogene (Figure 6).

## 3. Discussion

Differentiation of homologous chromosomes exists among genotypes of self-pollinated species. In *Triticum aestivum*, differences between homologs in their heterochromatin banding were documented [30]. Intraspecific polymorphism was mostly expressed for chromosomes of the B genome [31]. Due to chromosome-specific variation, the number of chiasmata and pairing of homologous chromosomes in F1 hybrids between cultivars was reduced compared to that of their parental inbred lines [32]. The observed recombination frequency between genetic markers depends upon the level of differences between the chromosomes of the parental cultivars involved in the cross [32,33].

Homologous chromosomes present in different wheat genotypes were called “heterohomologous” to differentiate them from euhomologous (veritably homologous) chromosomes present in a given genotype. The differences among heterohomologues are distributed along the chromosomes involved and are assumed to be due to differences in the non-coding portion of DNA sequences [34,35]. Deletions of small DNA sequences [36], the presence of different sets of transposable elements and variation in the number of their blocks located between genes [14,16,22,37], as well as SNP [38,39], may contribute to the differences between heterohomologous chromosomes.

The existence of several quite different variants of the DNA sequence at the *Gli-B1* locus of common wheat has been documented in our work. These variants could be designated, analogously to heterohomologous chromosomes, as “heteroallelic”, and alleles belonging to the same variant would be “homoallelic”.

RFLP seems to be especially useful for the comparison of allelic versions of a *Gli* locus, because restriction sites for a given endonuclease may occur at any location along the whole long gliadin locus, including expressed genes and non-coding sequences. Using the RFLP approach permitted several distinct variants of DNA sequences of the *Gli-B1* locus to be documented, strongly differing (without intermediate forms) in their restriction site positions. Moreover, some fine differences between alleles belonging to the same variant were revealed.

The length of the PCR products of the γ-gliadin SNP markers and sequencing the PCR products of the pseudogene *GAG56B* were used for more detailed analysis of two particular regions of the non-coding DNA inside the *Gli-B1* locus. In general, analysis of coding and non-coding parts of 15 allelic variants of the *Gli-B1* locus revealed at least seven “heteroallelic” variants of the *Gli-B1* locus (I–VII) without intermediate forms between them. Two pairs of allelic variants (II and III; IV and VII) were more closely related than the others. Nevertheless, allele *Gli-B1a* (variant VII, which was only documented in the genotype of the local variety Chinese-Spring), for example, differed from alleles of variant IV by the EM (APAGE) of the encoded γ-gliadin (Figure 1, Appendix A), by the RFLP-*Taq*I test (Figure 3a), by the length of the PCR product (Figure 4) and by the sequence of the pseudogene *GAG56B* (Figure 5 and Figure 6).

The allele *Gli-B1c* was identical to other alleles of group IV in the RFLP test (endonucleases *Taq*I and *Rsa*I) and in PLP and *GAG56B* tests but differed from them in another RFLP test (*Hae*III) and in APAGE. Therefore, we assume *Gli-B1c* is one more variant (VIII) related to group IV.

Earlier, six contrasting variants without intermediate forms were described for DNA sequences of the *Glu-B1* locus (B genome) among 58 cultivars of tetraploid and hexaploid wheat. It was suggested that these variants were present in genotypes of the diploid donors of this genome [13].

The pathways of origin of hexaploid wheat through successive crosses between diploid donors of the A and B genomes, and later, between AB and the D genome, are known in detail [40]. Naturally, heteroallelic variants of the *Gli-B1* locus might originate from different genotypes of the donor(s) of the B genome in hexaploid wheat, as suggested by [41], and inherit particularities of the *Gli-B1* locus of these donors. It is suggested that genotypes of the diploid donors of genomes A, B and D were already highly polymorphic before the origin of allopolyploids [42] and that crosses between wild tetraploids and young forms of hexaploid wheat [43] might also have contributed to the currently observed polymorphism of common wheat. A recent study showed great variation in common wheat germplasm (extensive structural rearrangements of DNA sequences and gene content) resulting from the history of wheat breeding aimed at improving different wheat characteristics, such as grain yield and quality, resistance to stresses and diseases, and adaptation to conditions of growth [44].

Analysis of the gliadin genotypes in about 1000 common wheat worldwide cultivars of the 20th century (Appendix A in [11]) showed that five main versions of heteroalleles (I, III, IV+VIII, V, VI) at the *Gli-B1* locus not only dominate in the germplasm studied but are also spatially structured around the world and probably adapted to regional eco-climatic environments [45]. We suggest that genetic diversity at the *Gli-B1* locus of common wheat germplasm studied is based mainly on the differences between variants of genotypes inherited by hexaploid wheat from its diploid donor(s) of the B genome. The donor(s) supplied common wheat with its own genetic variation of the distal part of chromosome 1B, which is now observed as heteroallelic variants at the *Gli-B1* locus. A considerable downturn in the level of genetic diversity of common wheat for this chromosome region would be caused only by the disappearance (impossible) of one of these variants.

If the level of genetic diversity of wheat germplasm of the 20th century calculated for the *Gli-B1* locus depended decisively upon several heteroallelic variants, it would not be as great as that calculated using all (more than 20 [11]) known individual alleles present in registered wheat cultivars. Indeed, several allelic members of one variant might have a mutational origin (for example, alleles of variant I) and not differ among them in any test applied in this work. Although the loss of rare alleles over time may be considered a display of genetic erosion [46], the disappearance of a rare allele (a member of one of the principal variants) would have only a minor influence on the level of genetic diversity.

## 4. Materials and Methods

### 4.1. Plant Materials

The majority of grain samples of the common wheat (*Triticum aestivum* L.) cultivars used in this work were obtained from genetic and/or breeding laboratories in their countries of origin and studied earlier (Appendix A in [11]). In total, 916 registered cultivars from 18 countries were considered, in which alleles at the *Gli-B1* locus were identified [11].

For this work, some grain samples not studied earlier were taken from local collections in Italy, Spain and Ukraine. Also included in the study were grain samples not studied earlier of several registered cultivars and important genotypes (for example, Chinese Spring), plus several Spanish landraces. The widest possible, first, allelic variation at the *Gli-B1* locus and, second, origin of cultivars were thereby covered.

Names of cultivars, country of origin, year of registration and spring/winter habit were taken from the CIMMYT GRES database [47].

### 4.2. Molecular Analysis

In the current work, some grain samples from local collections were analyzed with the aim of confirming their gliadin genotypes. To do this, an acid polyacrylamide gel electrophoresis (APAGE) routine procedure [11] was used. Two-dimensional electrophoresis (APAGE × SDS-electrophoresis, TDE) was performed as previously described [28].

For RFLP analysis, DNA was extracted from seven-day-old seedlings) using a modified CTAB procedure [48]. DNA digestion, gel electrophoresis, blotting and hybridization were carried out as described by Gebhardt et al. [29,49]. For the identification of the γ-gliadin DNA sequences, the probe K32 (the cDNA clone) specifically recognizing these sequences was applied [50].

DNA was extracted from dry single seeds using the CTAB method [51]. DNA qualities were evaluated on agarose gels, and DNA concentrations were determined spectrophotometrically using an ND-1000 instrument (Thermo Fisher Scientific, Wilmington, North Carolina, USA). All extractions were diluted to 100ng/µl and stored at −20 °C until being used.

PCR amplification was performed in the thermocycler Analytik Jena (Flex Cycler, Jena, Germany) using GliB1.1 and GliB1.2 primers [25]. Products of amplification for each cultivar (three or more single seeds per cultivar were studied) were fractionated in 7% polyacrylamide gels (280 V) for two hours and stained as proposed by the Silver sequence TM DNA Sequencing System Technical Manual (Promega, Madison, WI, USA). The lengths of the amplification products were compared in neighboring lanes of the same slab gel. The approximate length of the product of amplification for each cultivar was calculated using standard markers with fragments of known length.

γ-Gliadin gene *GAG56B* was amplified with GAG29 and GAG30 primers [24]. PCR-amplified fragments were purified with sepharose columns and sequenced by capillary electrophoresis at Macrogen (Macrogen Europe, Amsterdam, The Netherlands). Sequences were analyzed with Sequencher^®^ version 5.0 sequence analysis software (Gene Codes Corporation, Ann Arbor, MI, USA) and aligned with acn M13712 in order to detect polymorphisms. The information obtained was used to reconstruct *GAG56B* for each cultivar and to define haplotypes. DNA sequences were aligned using clustalW [52]. Phylogenetic trees from the nucleotide alignment were calculated employing the Neighbor-joining method, with Jukes-Centre distance and 100 bootstrap.

The trees of the relationship between alleles at the *Gli-B1* locus based on the RFLP data were also designed by the Neighbor-joining method.

## Figures and Tables

**Figure 1 ijms-22-01832-f001:**
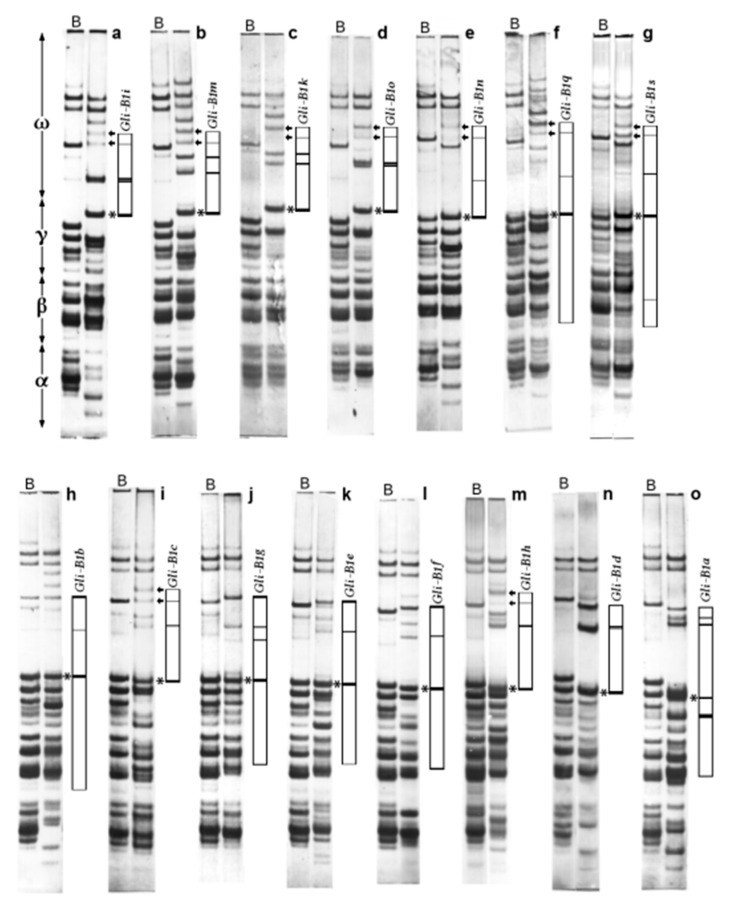
APAGE electrophoregrams of standard common wheat cultivars carrying different alleles at the *Gli-B1* locus. (**a**), Insignia (Australia); (**b**), Silvana (Romania); (**c**), Pyrotrix-28 (Kazakhstan); (**d**), Levent (Bulgaria); (**e**), Intensivnaya (Kirgizstan); (**f**), Lesostepka-75 (Ukraine); (**g**), Est-Mottin (Italy); (**h**), Marquis (Canada); (**i**), Siete-Cerros-66 (Mexico); (**j**), Galahad (UK); (**k**), Solo (Germany); (**l**), Ducat (France); (**m**), Barbilla-Leon (Spain); (**n**), Caprock (USA); (**o**), Chinese Spring (China). B, universal standard cultivar Bezostaya-1 (*Gli-B1b*) (Russia) [11]. The major γ-gliadin encoded at the *Gli-B1* locus (asterisk) and ω-gliadins encoded by the allele *Gli-B5b* (arrows) are indicated. α, γ and ω, mark the protion of the electrophonegrams where α-, γ- and ω-gliadins migrate respectively. The blocks of jointly inherited gliadin electrophoretic bands are shown schematically.

**Figure 2 ijms-22-01832-f002:**
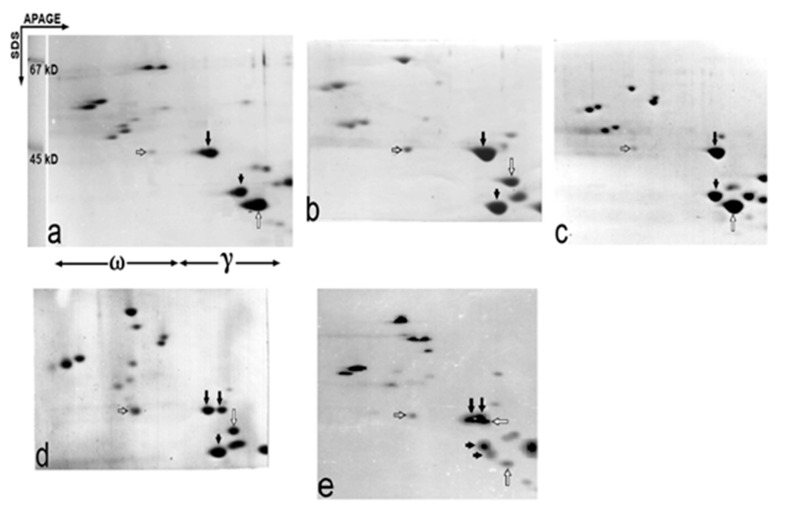
Two-dimensional (APAGE × SDS) separations of gliadin of common wheat cultivars and of F2 seeds of their crosses. (**a**), Skorospelka-U (*Gli-B1m*); (**b**), Tarasovskaya-29 (*Gli-B1b*); (**c**), Dneprovskaya-521 (*Gli-B1d*); (**d**), F2 seed heterozygous at the *Gli-B1* from the cross between cultivars Rusalka (*Gli-B1d*) and Bezostaya-1 (*Gli-B1b*); (**e**), F2 seed heterozygous at each of the *Gli-1* loci from the cross between Kharkovskaya-6 (*Gli-A1j*, *Gli-B1e*, *Gli-D1i*) and Skorospelka-U (biotype *Gli-A1m*, *Gli-B1m*, *Gli-D1c*). The long-tailed black arrows indicate the *Gli-B1*-encoded γ-gliadins, and the transparent short-tailed arrow shows the *Gli-A3*-controlled ω-gliadin having an MW of 41 kD (“internal marker”). The transparent long-tailed arrows, and the short-tailed black arrows, show the *Gli-A1*-encoded and the *Gli-D1*-encoded γ-gliadins, respectively. Only the γ- and ω-zones of the two-dimensional electrophoregram are shown.

**Figure 3 ijms-22-01832-f003:**
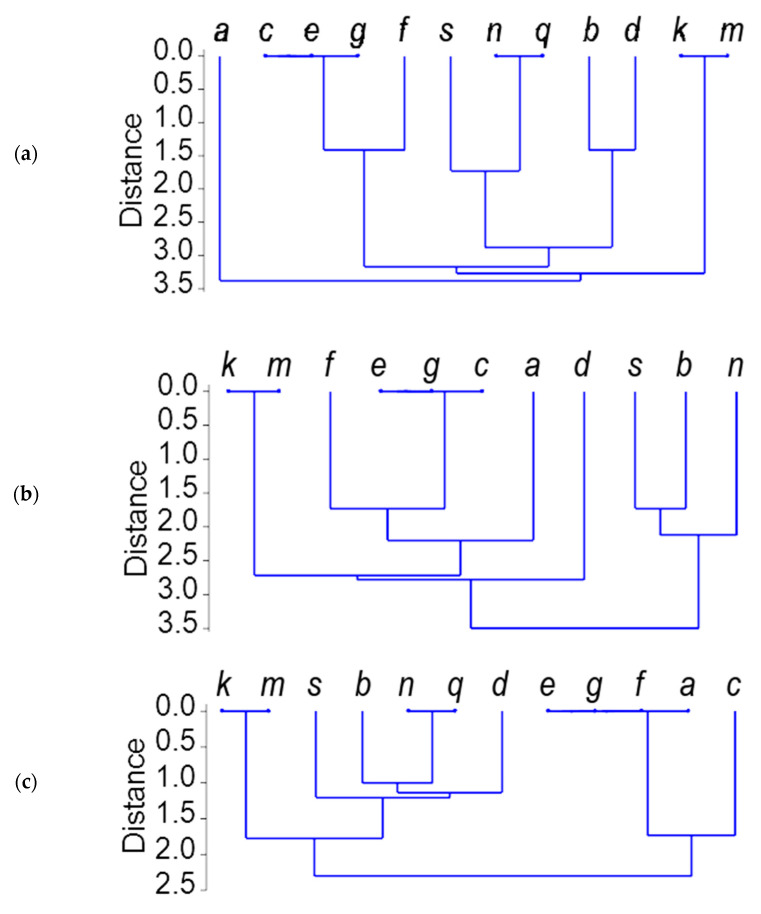
Relationships among alleles at the *Gli-B1* locus based on the restriction fragment length polymorphism (RFLP) data. (**a**), *Taq*I; (**b**), *Rsa*I; (**c**), *Hae*III.

**Figure 4 ijms-22-01832-f004:**
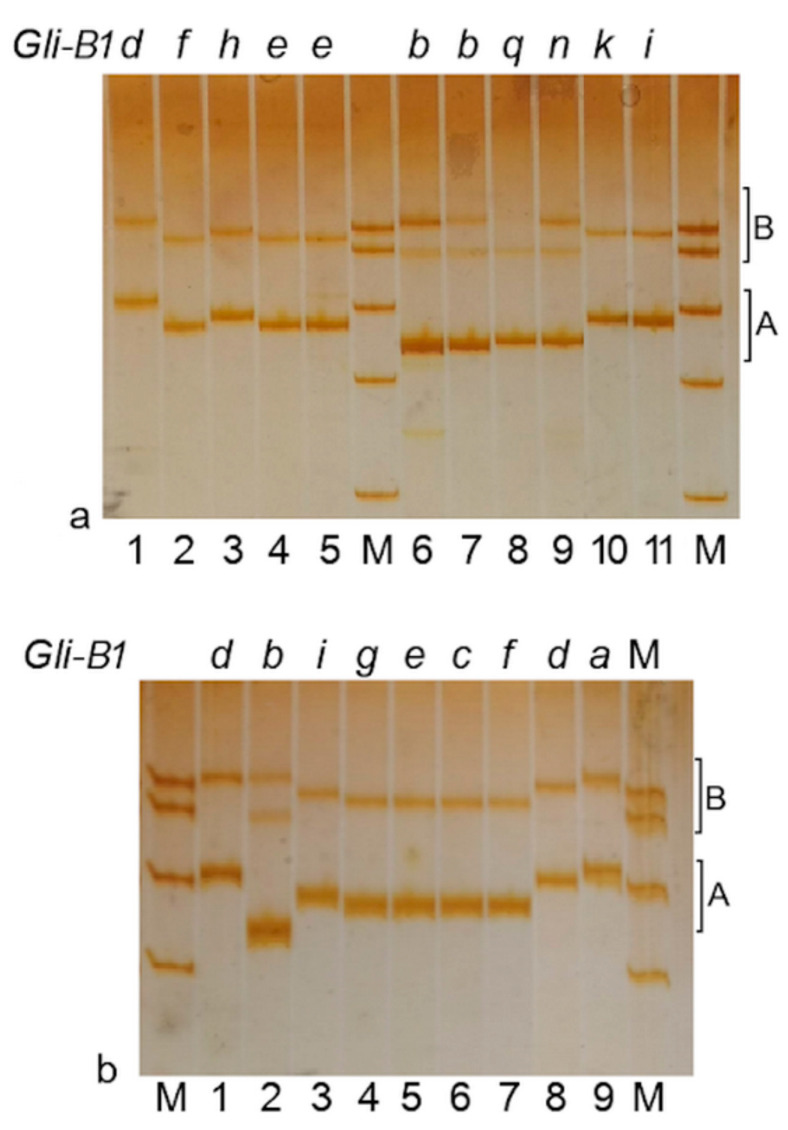
PCR (primers B1.1 or B1.2) Product Length Polymorphism (PLP) among alleles at the *Gli-B1* locus. A and B represent two differentiated zones in the gel. Alleles at the *Gli-B1* loci for each sample analyzed are indicated in the upper part of the gel. (**a**) 1, Pavon-F-76; 2, Cappelle-Desprez; 3, Ardec; 4, Escualo; 5, Glenlea; 6, Gabo; 7, Marquis; 8, Goelent; 9, Intensivnaya; 10, Mentana; 11, Insignia; 1–5, primers GliB1.2; 6–11, primers GliB1.1; (**b**) 1, Laura; 2, Bezostaya-1; 3, Insignia; 4, Galahad; 5, Escualo; 6, Prinqual; 7, Arminda; 8, Suneca; 9, Chinese-Spring; 1, 4–9 primers GliB1.2; 2, 3, primers GliB1.1. Alleles at the *Gli-B1* locus are indicated above the lanes. M, marker pUC19/*Msp*l with DNA fragments of length 501, 489, 404, 331 and 242 bp, respectively. The polymorphism of the PCR-derived DNA fragments of higher length (zone B) was not considered in this work.

**Figure 5 ijms-22-01832-f005:**
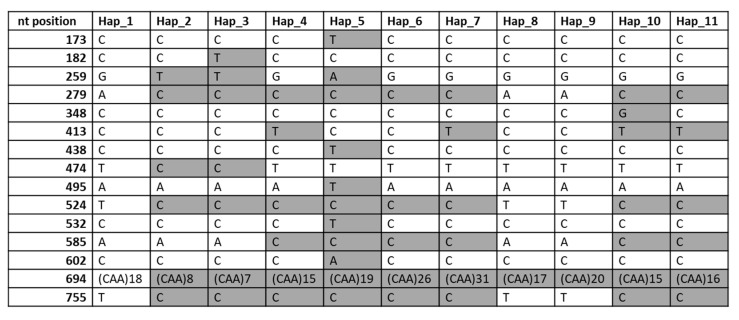
Differences in nucleotide sequence among 11 haplotypes of the pseudogene *GAG56B*. Differences with respect to haplotype 1 are marked in grey. Positions are counted from the beginning of the primer employed in the DNA amplification.

**Figure 6 ijms-22-01832-f006:**
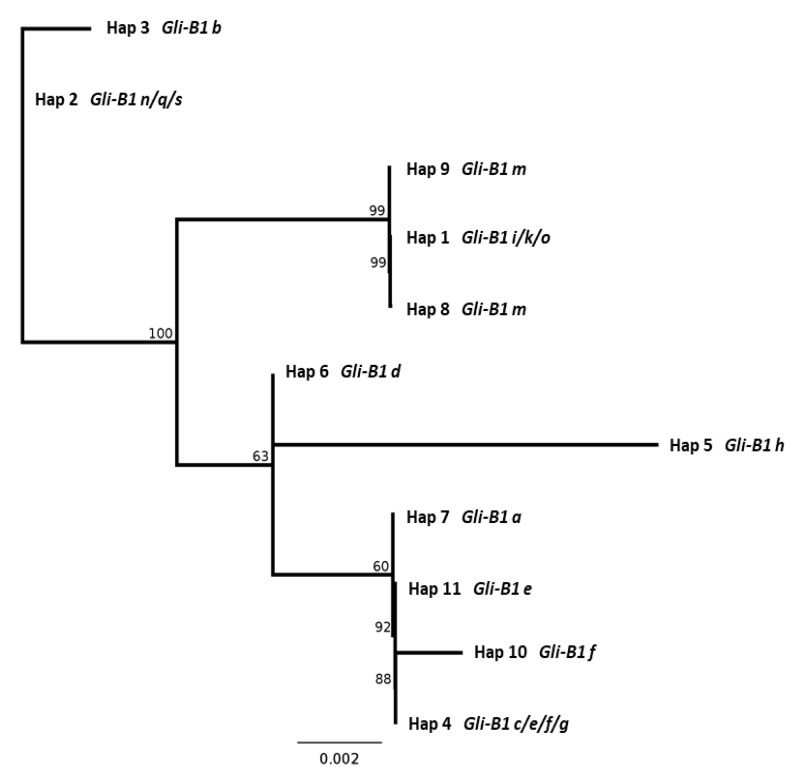
Relationships among allelic variants of the *Gli-B1* locus based upon the sequence of the pseudogene *GAG56B* located within the *Gli-B1* locus.

**Table 1 ijms-22-01832-t001:** Cultivars studied, their alleles at the *Gli-B1* locus, and the approximate length for the product obtained using GliB1.1 or GliB1.2 primers.

Cultivar	Country	Year	Allele	Primer	Length (bp)
Bezostaya-1	Russia	1959	*b*	GliB1.1	369
Gabo	Australia	1942	*b*	GliB1.1	369
Marquis	Canada	1907	*b*	GliB1.1	369
Insignia	Australia	1946	*i*	GliB1.1	400
Mentana	Italy	1913	*k*	GliB1.1	400
Titien	France	1985	*m*	GliB1.1	400
Aragon-03	Spain	1940	*o*	GliB1.1	400
Intensivnaya	Kyrgyzstan	1978	*n*	GliB1.1	372
Goelent	France	1985	*q*	GliB1.1	372
Salmone	Italy	1980	*s*	GliB1.1	372
Cartaya	Spain	1984	*l*	GliB1.1 or GliB1.2	-
Chinese-Spring	China	-	*a*	GliB1.2	415
Pavon-F-76	Mexico	1971	*d*	GliB1.2	409
Laura	Canada	1986	*d*	GliB1.2	409
Rinconada	Spain	1981	*d*	GliB1.2	409
Suneca	Australia	1981	*d*	GliB1.2	409
Prinqual	France	1978	*c*	GliB1.2	397
Escualo	Spain	1981	*e*	GliB1.2	397
Glenlea	Canada	1972	*e*	GliB1.2	397
Arminda	Netherlands	1976	*f*	GliB1.2	397
Cappelle-Desprez	France	1946	*f*	GliB1.2	397
Galahad	UK	1983	*g*	GliB1.2	397
Ardec	Belgium	1979	*h*	GliB1.2	402
Caia	Portugal	-	*h*	GliB1.2	403

**Table 2 ijms-22-01832-t002:** Variants of DNA sequences (haplotypes) of the pseudogene *GAG56B* in cultivars having different alleles at the *Gli-B1* locus.

Cultivar	Country	Year	Allele	Haplotype (Figure 5)	EM of γ-Gliadin ^1^
Insignia	Australia	1946	*i*	1	I
Pane-247	Spain	1960	*k*	1	I
Etoile-de-Choisy	France	1950	*m*	1	I
Aragon-03	Spain	1940	*o*	1	I
Intensivnaya	Kyrgyzstan	1978	*n*	2	II
Spada	Italy	1985	*n*	2	II
Lesostepka-75	Ukraine	1945	*q*	2	II
Salmone	Italy	1980	*s*	2*	II
Alcalá	Spain	1984	*b*	3	III
Anza	Mexico, USA	1971	*b*	3	III
Glenlea	Canada	1972	*e*	4	IV
Marius	France	1976	*f*	4	IV
Pernel	France	1983	*f*	4	IV
Adalid	Spain	1987	*g*	4	IV
Calodine	Italy	1991	*g*	4	IV
Diego	Spain	-	*c*	4	VIII
Prinqual	France	1978	*c*	4	VIII
Siete-Cerros-66	Mexico	1966	*c*	4	VIII
Ardec	Belgium	1979	*h*	5	V
Caia	Portugal	-	*h*	5	V
Pepital	Netherlands	1989	*h*	5	V
Cajeme-71	Mexico	1971	*d*	6	VI
Chopin	France	1984	*d*	6	VI
Katepwa	Canada	1981	*d*	6	VI
Chinese-Spring	China	-	*a*	7	VII
Pyrotrix-28	Kazakhstan	1973	*m*	8	I
Titien	France	1985	*m*	9	I
Astral	France	1972	*f*	10	IV
Floreal	France	1984	*f*	10	V
Saratovskaya-39	Russia	1968	*e*	11	IV

^1^ Variant of the electrophoretic mobility (APAGE) of the *Gli-B1*-encoded γ-gliadin (variant I, the slowest; variant VII, the fastest).

## Data Availability

Not applicable.

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
