# Peer review of "Heteroalleles in Common Wheat: Multiple Differences between Allelic Variants of the Gli-B1 Locus"

_ijms, 2021, doi:10.3390/ijms22041832_

Round 1

Reviewer 1 Report

The manuscript entitled "Heteroalleles in common wheat: …. the Gli-B1 locus" has novel information. The manuscript was well written and organized the huge data and the flow in a great way.

Minor comments to authors:

Please label a, b, and c close to the panel in Figure 3.

Improve the conclusions, not covered the entire results shown in this work.

T. aestivum should be italic throughout the manuscript.

**************

Author Response

We thank the reviewer for his/her remarks and we have modified the manuscript according to them:

  1. Figure 3 has been edited following the reviewer instructions, plus we have put the name of the different GliB1 alleles in italic.
  2. We have carefully edited our discussion section to avoid spelling and grammatical mistakes. We believe with those changes the conclusions are clearer.
  3. When formatting the manuscript we lost all the italics along the text, this has been solved, now the names of the species, genes and alleles are all in italic.

We provide a changes tracked version of the manuscript. Moreover, we have re-submitted the supplementary material as changes on the reference list have been done.

Reviewer 2 Report

This is the interesting study about multiple differences between allelic variants of the gli-b1 locus.

The study updates us with the information about independent origin of at least seven variants of the Gli-B1 locus that might originate from deeply diverged genotypes of the donor(s) of the B genome in hexaploid wheat and therefore might be called "heteroallelic". The donor's particularities of the Gli-B1 locus might be conserved since that time, and decisively contribute to the current high genetic diversity of common wheat.

The readability of the manuscript is fluent and understandable. There are some minor concerns which should be addressed before accepting for publication:

  1. Additional English editing should be done; please check the use of prepositions (I am not a native English speaker but there might be some corrections needed for the final tuning of the manuscript).
  2. Please check and correct the English spelling and typing mistakes (eg. Table 1, line 233, bp instead of pb) within the whole MS
  3. Genus and species denominations are written in italics. Please check and correct in key words and the whole MS (e.g. in line 375).
  4. Within Introduction section please insert (lines 37, 38) a short description about gliadin importance in wheat yield in its products (in association with gluten?)
  5. Include concern no. 4. also in Discussion section.
  6. Please extend the MM section; the methods are poorly described. Please be a bit more specific (specify the DNA QC characteristics if checked; crucial RFLP steps (even it was generally cited);…)
  7. Please include same basic statistics/variability parameters within results section (e.g. informativity of the markers; add any other than only clustering method (e.g. PCA/PCoA).
  8. Please check the references (style and correct citation).

Author Response

We thank the reviewer for his/her review and we have modified the manuscript according to his/her suggested changes:

  1. One of the co-authors Dr. Rogers is an English native speaker, he has carefully edited the manuscript and corrected several spelling and grammatical errors.
  2. The small mistakes detected along the manuscript have been corrected.
  3. When formatting the manuscript we lost all the italics along the text, this has been solved, now the names of the species, genes and alleles are all in italic.
  4. The introduction has been modified, now it has two new paragraphs stating the importance of wheat and wheat proteins (lines 36-55).
  5. Wheat yield is controlled by a big set of genes and might be negatively correlated with protein content but not with the alleles of these proteins. Regarding to the relation with gluten obviously gliadin alleles might affect gluten index and other parameters. However in this work we are not performing phenotypic evaluations. Actually we would like to highlight the other "use" for wheat proteins as genetic markers related with cultivars identification and evolution which is the focus of the article.
  6. Material and methods section has been improved. We have added details about the criteria to select the analyzed cultivars (lines 453-454), and RFLPs and DNA QC (lines 462-470).
  7. Regarding to the new analysis proposed by the reviewer, we would like to clarify that PCA is based on quantitative data and ours are qualitative. PCoA could be done in order to put together the data from all the different methodologies and represent the distance among the different cultivars. However, this is not what we intended. We did not want to compare the cultivars, we wanted to detect differences inside the GliB1 locus, and we cannot do that if we put all the data together. We agree with the reviewer that employing different methodologies will give a more general picture. That is why we have studies the GliB1 locus with different approaches. Finally, we would like to remark that our intent is not to cluster the cultivars, but to detect phylogenetic relations among the different GliB1 alleles and this is best done with a phylogenetic tree.
  8. As we have included new references all of them have been re-numbered in the text and we have also checked and changed the reference list.

We provide a changes tracked version of the manuscript. Moreover, we have re-submitted the supplementary material as changes on the reference list have been done.

Reviewer 3 Report

Technical the paper is very good with nicely presented results however I do have some concerns:

HETEROALLELES IN COMMON WHEAT:  MULTIPLE DIFFERENCES BETWEEN ALLELIC VARIANTS OF THE Gli-B1 LOCUS

Perhaps the title could be:

THE Gli-B1 LOCUS IN COMMON WHEAT SHOWS MULTIPLE DIFFERENCES BETWEEN ALLELIC VARIANTS

Introduction:

The authors need to rewrite and restructure the introduction. They have to set the scene for the reader: They need to introduce wheat as one of the three main crops globally, the importance of wheat including global yields etc.,

They should also introduce the importance of the wheat grain and the endosperm bringing in the importance of glutenin and gliadin as the major protein network within the wheat grain and some background information on their function and relevance and then begin their focus on gliadins.

Also, what is their biological question? Is this work done as part of a breeding programme to improve grain quality by studying the variation in a range of wheat cultivars? This needs to be defined more clearly in the final paragraph.

Where the cultivars used in this study chosen at random or are, they elite cultivars in their respective countries? Why was this subset of cultivars chosen?

I think it would have been better to have germinated the seeds and extract DNA from single plants using the CTAB method.

The Materials and Methods section is a little thin and references are used- did all methods transfer over or did you have to modify any of the methods used in this study?

Author Response

We thank the reviewer by his/her suggestions and have modified the manuscript according to them.

  1. We have added two new paragraph in the introduction, the FAOSTAT data related to wheat are presented in order to highlight the importance of this species (lines 36-42). Also we have included the key unresolved questions we want to answer (lines 50-55). Moreover, we have carefully edited the final paragraph in the introduction to make it more understandable.
  2. The set of cultivars was chosen to represent the allelic diversity of the GliB1 locus, maximizing the geographic range. We have included those criteria in the material and methods section (lines 453-454).

We provide a changes tracked version of the manuscript. Moreover, we have re-submitted the supplementary material as changes on the reference list have been done.

Round 2

Reviewer 3 Report

I would like to thank the authors for taking my recommendations on board but i still feel the biological question and reason for carrying out this work needs to be clarified in the last paragraph of the introduction.

Also- line 37- use tonnes 

Line 44- it is major not mayor

Author Response

We thank the reviewer for detecting two small typos that have now been corrected.

Besides, based on his/her comments we have added a summary of the most relevant information presented in the introduction, defined the question we aim to analyze in our work and the reason it will have a direct impact in wheat breeding (lines 103-110). We hope this new version will be suitable for publication.
